# Study of the Operation Process of the E-Commerce Oriented Ecosystem of 5Ge Base Station, Which Supports the Functioning of Independent Virtual Network Segments

Viacheslav Kovtun [1] and Ivan Izonin [2,*]

1 Department of Computer Control Systems, Vinnytsia National Technical University, Khmelnitske Shose, 95, 21000 Vinnytsia, Ukraine; kovtun_v_v@vntu.edu.ua
2 Department of Artificial Intelligence, Lviv Polytechnic National University, Stepan Bandera Str., 12, 79013 Lviv, Ukraine
* Correspondence: ivan.v.izonin@lpnu.ua

**Abstract:** According to specifications, flexible services for traffic management should be implemented within the 5G platform in order to improve its efficiency, which is and will remain an actual task. For the first time, the article presented here proposes a mathematical model for the operation process of an e-commerce-oriented ecosystem of a 5Ge base station, the information environment of which supports the operation of independent virtual network segments that provide terminal–segment information interaction services. In contrast to existing models, the presented model describes the studied process as a multi-pipeline queuing system, the inputs of which are coordinated with the flows of requests for communication with the relevant virtual network segments. The distribution of the total resources between the weighted virtual network segments in the simulated system is dynamically conducted by the appropriate software control mechanism. It considers the address intensities of new incoming requests and the maintenance of received incoming requests, but throughout the scale of the information environment of the 5Ge base station ecosystem. Based on the created mathematical model, a functional algorithm for the forced termination of an active terminal–segment information interaction session in the overloaded virtual network segment and the control mechanism of the distribution of the released system resources between other virtual network segments that takes into account the degree of their overload are formulated. The simulation and computational experiments showed that the implemented forced termination algorithm and system resource management mechanism allow the 5Ge base station to continue receiving incoming requests despite the overload of individual virtual network segments. It is empirically shown that the proposed services are effectively scaled concerning the value that is generally available for the distribution of the number of system resources and the allocation method within the guaranteed amounts of system resources for individual virtual network segments.

**Keywords:** e-commerce; 5Ge ecosystem; virtual network segment; mathematical model; availability; terminal-segment interaction session; resource allocation management mechanism

## 1. Introduction

Over the last decade, the networks and communication systems industry has seen an exponential increase in both the number of subscriber devices (intended, including for e-commerce) and the total amount of traffic transmitted by wireless channels in general-purpose communication networks. According to various sources, this trend will continue [1–3]. An increasing proportion of subscriber devices are mobile devices, which are already allocated to a specific ecosystem known as the Internet of Wearable Things (IoWT) [3]. The list of IoWT device classes is constantly expanding and includes traditional smartphones and smartwatches, bracelets, augmented and virtual reality glasses, etc. IoWT

devices in the conditions of modern megacities generate the lion's share of information, causing an extremely high load on the subscriber segment of communication networks.

In addition to the constant growth of information interaction, mobile traffic is affected by disruptive factors in current conditions, the appearance of which is only stochastically localized in time and space. In terms of time, the intensity of mobile traffic is distributed unevenly throughout the day, which makes it necessary to activate the appropriate services to balance the network operation communication process. At the same time, the space factor manifests itself in the phenomena of interference or the blocking of some subscribers to others in the conditions of complex urban relief of modern cities. Compensating for these factors is becoming an increasingly difficult task as the number of IoWT devices increases.

Both network equipment manufacturers and standardization organizations are constantly looking for symmetrical responses to current challenges that accompany the expansion of the infosphere. Additionally, the modern mobile networks of the 4G-LTE generation are based on an effective basis of physical and channel levels technologies. In addition, it is already close to the theoretical upper limit of the bandwidth of wireless communication channels. Therefore, a promising direction that will further increase the information capacity of wireless communication channels is an expansion of the range of available frequencies. However, due to the load of the electromagnetic spectrum at frequencies below 6 GHz, advanced communication systems should focus on operation at the frequency range of 30–300 GHz. According to the leading organization for the standardization of mobile communication systems, 3GPP, devices operating in frequency band 30–100 GHz wireless radio access systems already belong to the 5G generation. In particular, the characteristic features of such systems are a slight delay on the wireless interface and a peak data rate of up to 10 Gbps. These requirements are provided by appropriate connection control, which, according to the specifications of the New Radio standard, was entrusted by base stations that support 5Ge technology.

Due to the use of the millimeter frequency range, 5G generation communication devices use antenna arrays characterized by high directivity. This technological solution provides high bandwidth communication channels and significantly reduces the interference created by such systems. However, the practice of operating 4G-LTE and 5G communication devices has revealed some unique negative phenomena. In particular, the physical blocking of the focused communication channel leads to a frequent and prolonged drop in the signal level, which cannot be compensated for by the technologies used in the 2G and 3G generations. Depending on the degree and duration of the drop phenomenon at the signal level, its appearance can cause both an abnormal increase in the number of resources directed by the base station to maintain a given speed of information interaction and the premature termination of the information interaction process. Thus, to improve subscriber service quality in 5G generation networks, it is necessary to solve many new tasks that can only be solved by creating a methodology of mathematical models to adequately describe the processes that are specific to this generation networks.

The motivation for the current research is the formalization of information technology in the composition of models, methods, and algorithms to control the information interaction process of a set of terminals and base stations in the 5Ge ecosystem that are focused on applications in e-commerce. The hierarchy of the applied tasks that allow this motivation to be achieved is formulated in Section 3.1. This article represents the first published work by the current authors in this specific area.

Thus, the research object is the operation process of the e-commerce-oriented ecosystem of the base station 5Ge, which supports the functioning of independent virtual network segments.

The main contributions of this paper can be summarized as follows:

1. We have created a mathematical model for the operation process of the e-commerce-oriented ecosystem of the 5Ge base station, the information environment of which supports the operation of independent virtual network segments that provide terminal–segment information interaction services;

2. We have designed a functional algorithm for the forced termination of an active terminal–segment information interaction section in an overloaded virtual network segment;

3. We have formulated the control mechanism of the distribution of the released system resources between other virtual network segments by taking into account the degree of their overload;

4. We conducted simulation and computational experiments showing that the implemented forced termination algorithm and system resource management mechanism allow the 5Ge base station to continue receiving incoming requests despite the overload of individual virtual network segments.

The structure of the article is as follows: 1. The Section 2 provides a critical analysis of the state of the research related to the topic discussed in the present article. The results of the analysis revealed an area that has so far been ignored in general studies. The subject of the research was also selected in the identified area; 2. In Section 3, the research objectives are set, and the proposed information technology for describing the process of information interaction in the e-commerce-oriented ecosystem of the 5Ge base station is theoretically substantiated; 3. The Sections 4 and 5 are devoted to the simulation modeling of the presented information technology and the analysis of the adequacy of the results demonstrated by it; 4. In Section 6, the conclusions concerning the urgency of the chosen theme and the received theoretical and applied results are formulated. Directions for future research are also determined.

## 2. State-of-the-Art

The task of 5G generation communication networks is to meet the growing needs of states, institutions, businesses, and individuals in mobile communications of the appropriate level of quality. It is assumed [2,4–6] that the 5G generation networks will play a key role in transforming megacities into "Smart Cities". Smart Cities provide the socio-economic benefits provided by the advanced digital economy to both citizen and society, with the intensive target using data.

The concept of building 5G communication networks was created based on improving the quality, reliability, and productivity of end-users by offering new services and applications with gigabit data rates. In particular, by raising the level of deployed mobile networks to the 5G generation, mobile operators will provide 3G and 4G generation services and will develop their solutions and services for consumers and industry. They will be implemented in independent virtual network segments at the level of one base station and their conglomerate [7–10].

Although the standardization of 5G generation networks is still ongoing, 5G commercial networks have been rolling out since 2020. The GSM Association predicts that by 2025, the number of one-time sessions to 5G networks will exceed $1.1 \cdot 10^9$, about 12% of the total expected number of events of this type [11–13]. It is also projected that by 2025, the total profits of mobile operators will increase by about 2.5% and reach the level of USD $1.3 \cdot 10^{12}$ [12,14].

From a technical point of view, the transition of communication networks to the 5G generation level will be accompanied by a significant increase in speed and a reduction of communication delays. In particular, 5G generation communication networks should function so that the communication delay does not exceed 1 ms on the wireless segment of the signal distribution channel, which allows the implementation of critical application services based on these networks [15–18]. Additionally, in 5G communication networks, a peak speed of up to 10 Gbps is technologically possible for the end-user, allowing operators to provide a wide range of high-speed broadband services and allowing it to become a natural alternative to satellite communications.

Mobile operators, equipment manufacturers, and network standardization organizations have identified the three most promising and fundamentally different categories of services implemented in 5G generation networks. These are the Enhanced Mobile Broad-

band (eMBB) service [19–21], the Ultra-Reliable Low-Latency Communications (URLLC) service [22–24], and the Massive Machine-Type Communications (mMTC) service [25–27].

The eMBB service aims to meet the needs of ultra-high-speed in the subscriber segment [26–29]. The need for such fast speeds has arisen in users of applications for watching ultra-high-definition video; using goggles and helmets for augmented and virtual reality; and the transmission and operation of "Big Data." It is expected that eMBB will be the main application of 5G at the early stages of its deployment. Some mobile operators see eMBB as a promising solution to the "Last Mile" problem for areas with no modern cable network infrastructure [28].

The high-quality level of latency and cybersecurity inherent for the URLLC implemented on the 5G platform will play an essential role in developing intelligent transport systems of the future, allowing vehicles to switch between themselves and to create new services for self-driving cars and trucks [29,30]. The need for a URLLC becomes apparent in the following example: a self-driving vehicle controlled by a cloud driving service must stop, accelerate, or turn immediately, according to real-time commands. Any delay or loss of connection in the information interaction of such a stand-alone car with the base station can lead to catastrophic consequences. The low latency inherent in the URLLC also allows us to consider the 5G generation network as a basis for implementing services for remote surgery, industrial automation, and real-time finance process control [31,32].

It is expected that with the mMTC service, 5G generation networks will also contribute to the development of smart cities and the Internet of Things by deploying sensor networks in cities and rural areas. The underpinnings of the 5G platform for security and reliability of information interaction will make such networks suitable for use in critical services [33–38], such as police, security, energy, water supply, and health services.

The range of potentially promising services that can be implemented precisely through the capabilities of 5G generation communication networks is not limited to the mentioned eMBB, URLLC, and mMTC. These services only represent the most typical real-world scenarios that have polar requirements for the quality characteristics of communication services. These good services form the base, which, in Figure 1, is presented in a triangle [17]. The sides of the triangle symbolize the fixed quality requirements for access speed, transmission reliability, and delay time. Some applications can be implemented both inside the triangle and on its faces by combining the appropriate set of these quality characteristics.

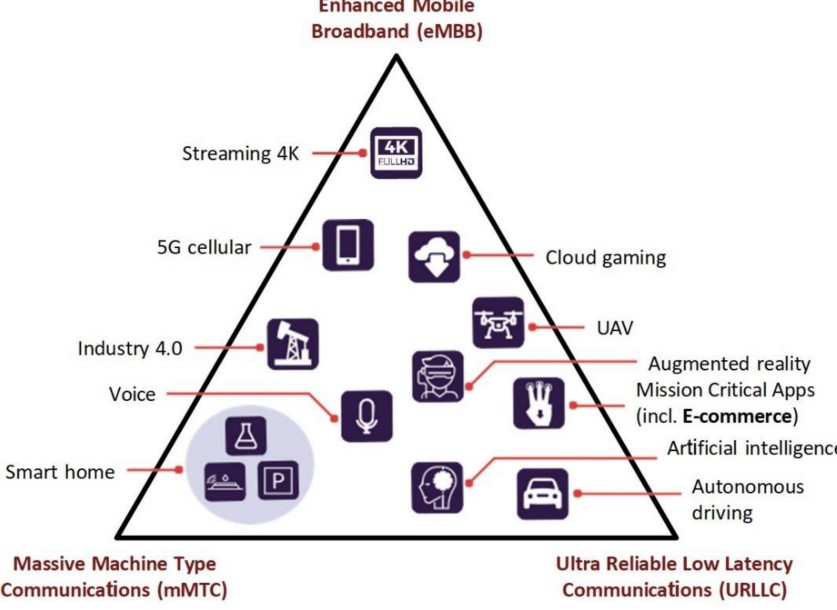

**Figure 1.** Services that can be implemented on the technological basis of 5G generation communication networks [17].

As mentioned in Section 1, the 5Ge platform implements technologies for forming independent virtual network segments in the information environment of the base station [7–10]. To support each such elements, system resources are allocated, and errors or malfunctions in one segment do not affect the Quality of Service (QoS) value for other components. In other words, the isolation of the virtual network segment to provide the appropriate service nomenclature of acceptable quality is guaranteed. At the same time, the algorithm for radio resource allocation should provide the effective targeted use of the system resources of the base station by taking the fact that each active virtual network segment is allocated a corresponding guaranteed amount of these resources into account [8,10].

This topic has recently attracted the attention of researchers. In [7–9], a flexible model for structuring a radio access network (RAN) is presented. The model focuses on providing an appropriate level of isolation for the resources of Virtual Network Operators (VNOs), which are network traffic tenants. The isolation level is defined to ensure that the Service Level Agreement (SLA) between the VNOs is not violated, even if different network settings change. The desired isolation level is determined by optimizing the RAN infrastructure used by dynamically allocating radio resources between different segments fairly.

Articles [10,39] also consider virtual radio resource management (VRRM), which provides the optimal use of virtualized resources of the infrastructure provider between several virtual network operators. The articles present the architecture of the VRRM modeling tool in terms of queuing systems [39]. With the help of the developed tool, the analysis of the practical scenario with three suppliers and different types of SLA can be conducted, and productivity indicators at a change loading site on traffic and SLA can be investigated.

In studies [11,15,40], the theoretical basis for the multi-operator scheduling (MOS) distribution of radio resources is considered. Due to the dynamic adaptation to the load of the communication channel, the centralized approach maximizes spectral efficiency for several operators with complete control over the guarantee of sharing.

However, something that is common to all of the proposed studies is to ignore the actual congestion scenario of the individual virtual network segments during the operation of the base station. In these models, if the incoming request is sent to the virtual network segment, the guaranteed amount of resources allocated by the base station is exhausted. Such an incoming request is lost even if free system resources are available in the information environment of the base station. To model this aspect of the 5Ge base station operation process, the authors propose choosing the mathematical apparatus for queuing systems, mathematical and simulation modeling, and experimental planning theory as the ***subject*** of research.

## 3. Materials and Methods

### 3.1. Statement of Research

The information provided in Section 2 confirms the relevance of the distribution of the communication resources of the cellular communication base station between the elements of the set of selected logical virtual network segments. A characteristic feature of this radio resource allocation technology is the independence of virtual network segments both at the computing resource frequency level and at the active service level. Independence means that for a given base station, the values of service quality indicators in a particular virtual network segment are not affected by the operation of the other virtual network segments.

The implementation quality of such network virtualization technology is entirely determined by the efficiency of the target control software mechanism, which operates at the level of the information environment of the base station. The creation of such a mechanism is only possible if there is an adequate profile for a mathematical model. Therefore, the present study aims to create a tool to manage the operation of the e-commerce-oriented ecosystem of the 5Ge base station, the information environment of which supports the functioning of independent virtual network segments.

Let us investigate the operation process of the localized e-commerce-oriented ecosystem of the 5Ge base station, the information environment of which supports the functioning of $S$ virtual network segments, provided that the control mechanism distributes the $C$ c.u. of the systemic recourses between them. According to the adopted QoS policy, each $s$-th $s \in S$ virtual network segment must support information terminal–segment interaction with a speed whose value is in the range of $[a_s, b_s]$, $a_s, b_s \geq 0$, $a_s \leq b_s$. The actual value of the rate is a function of the time and number of active sessions of terminal–segment interaction on the scale of the entire information environment of the base station. On the scale of a single virtual network segment, available system resources are evenly distributed between active terminal–segment interaction sessions.

We present the concept that was just described as a multi-pipeline queuing system, the inputs of which are consistent with the $S$ flows of requests to establish terminal–segment interaction with $S$ virtual network segments, respectively. All of these flows are Poisson with the intensities $\lambda_s$ $s \in S$. All submissions include service information about the desired amount of system resources. The values of these stochastic quantities are distributed according to the exponential law with the parameter $1/\mu_s$ $s \in S$. To support the incoming request received by the $s$-th virtual network segment, the control mechanism allocates a finite amount of system resources $r_s \in [a_s, b_s]$ that determines the actual speed of the address of the terminal–segment information interaction. Thus, the total volume of the system resources that is available creates $C$ c.u. At each initiation or termination of a terminal–segment information interaction session, the control mechanism redistributes the system resources between the virtual network segments.

Thus, the objectives of the research are:

1. To formalize a mathematical model for the operation process of the e-commerce-oriented ecosystem of the 5Ge base station, the information environment that supports the functioning of independent virtual network segments, in the queuing theory paradigm;
2. To formulate a functional algorithm for the forced termination of an active terminal–segment information interaction session in the overloaded virtual network segment and the management mechanism needed for the distribution of the released system resources between other virtual network by segments based on the aforementioned mathematical model by taking into account their overload level;
3. To conduct simulation and computational experiments to prove the adequacy of the created mathematical model and the effectiveness of available services made on its basis.

### 3.2. Mathematical Model of the Process of Operation of E-Commerce Oriented Ecosystem of 5Ge Base Station, the Information Environment of Which Supports the Functioning of Virtual Network Segments

We identified the operation of the localized e-commerce-oriented ecosystem of the 5Ge base station, the information environment that supports the operation of $S$ virtual network segments, as a stochastic function $X(t) = \{m_s(t)\}$ $s = \overline{1, S}$, where $m_s$ is the number of active terminal–segment information interaction sessions for the $s$-th virtual network segment at the time $t$: $m_s \in \left\{0, 1, \ldots, \left\lceil \frac{C}{a_s} \right\rceil\right\} \forall s \in S$. Let us define the state space of the stochastic process $X(t)$: $X = \left\{(m_1, m_2, \ldots, m_s), \sum_{s=1}^{S} m_s a_s \leq C\right\}$. Suppose the control mechanism allocated system resources to support one active terminal–segment information interaction session for the $s$-th virtual network segment, which is $r_s$ c.u. Then, the intensity of the request service is equal to $r_s \mu_s \forall s \in S$.

To localize the $s$-th virtual network segment in the information space of the e-commerce-oriented ecosystem of the 5Ge base station, we make the following assumption: The control mechanism is guaranteed to allocate to its support $\overline{R}_s$ c.u. of the system resources, which, accordingly, guarantees the support of the $\overline{M}_s = \lfloor \overline{R}_s / a_s \rfloor$ active terminal–segment information interaction sessions. A virtual network segment in the information space from

which more active terminal–segment information interaction sessions are supported than their guaranteed supported number $\overline{M}_s$, will be denoted as being overloaded. In conditions where there is a shortage of free system resources, the support for guaranteed terminal–segment information interaction sessions in any virtual network segment is conducted due to premature the termination of the required number of active terminal–segment information interaction sessions in the overloaded virtual network segments. If a shortage of free system resources is observed in the absence of crowded virtual network segments, then the incoming request is lost.

Let us investigate the dynamics of the stochastic process $X$ in time: $X(t)$. Let the e-commerce-oriented ecosystem of the 5Ge base station be in the state $(m_1, m_2, \ldots, m_s) \in X$. Then, upon receipt of the $s$-th virtual network segment of the incoming request, one of the following scenarios is implemented:

1.  $(m_1, m_2, \ldots, m_s + 1, \ldots, m_S) \in X$—in the e-commerce-oriented ecosystem of the 5Ge base station, where there are enough free system resources; therefore, the input request is accepted, and the stochastic process $X(t)$ goes to state $(m_1, m_2, \ldots, m_s + 1, \ldots, m_S)$;
2.  $(m_1, m_2, \ldots, m_s + 1, \ldots, m_S) \neq X$—in the e-commerce-oriented ecosystem of the 5Ge base station, where there are not enough free system resources, therefore:

    (1).  If $m_s < \overline{M}_s$ and $m_k > \overline{M}_k$, $s, k \in S$, and $s \neq k$, then the system resources are released by forcibly terminating the active terminal–segment information interaction sessions in the $k$-th overloaded virtual network segment:
    $$B_s = \{(m_1, m_2, \ldots, m_S)\} : [(m_1, m_2, \ldots, m_s + 1, \ldots, m_S) \neq X] \cap$$
    $$\cap [(m_s < \overline{M}_s) \cup (m_k > \overline{M}_k \forall s, k \in S, s \neq k)]$$
    , where $B_s$ is the state space with forced termination of the active terminal–segment information interaction session. The functional realization algorithm from step 2.1 will be formulated further in Section 3.3;

    (2).  The input request is lost:
    $$D_s = \{(m_1, m_2, \ldots, m_s + 1, \ldots, m_S)\} : [(m_1, m_2, \ldots, m_s + 1, \ldots, m_S) \neq X] \cap$$
    $$\cap [(m_s \geq \overline{M}_s) \cup (m_k \leq \overline{M}_k \forall s, k \in S, s \neq k)]$$
    , where $D_s$ is the state space of the incoming request loss.

### 3.3. Functional Algorithm of Forced Termination of an Active Session of Terminal-Segment Information Interaction

Considering the high probability of a situation in which the e-commerce-oriented ecosystem of the 5Ge base station will be more than one overloaded virtual network segment, it is advisable to enter a parameter that will quantify the degree of their relative level of overload. We formalize the options to determine this characteristic parameter:

$$w_s^{(1)} = \begin{cases} 1 \forall m_s \leq \overline{M}_s, \\ \frac{1}{m_s - \overline{M}_s + 1} \forall m_s > \overline{M}_s; \end{cases} \tag{1}$$

$$w_s^{(2)} = \begin{cases} 1 \forall m_s \leq \overline{M}_s, \\ \frac{\overline{M}_s}{m_s} \forall m_s > \overline{M}_s; \end{cases} \tag{2}$$

$$w_s^{(3)} = \begin{cases} 1 \forall m_s \leq \overline{M}_s, \\ w = const \forall m_s > \overline{M}_s. \end{cases} \tag{3}$$

The need for the forced termination of an active terminal–segment information interaction session arises when the incoming request arrives at an unoverloaded virtual network segment $s$ and when the e-commerce-oriented ecosystem of the 5Ge base station lacks the required amount of free system resources: $\{[(m_1, m_2, \ldots, m_s + 1, \ldots, m_S) \neq X] : (m_s + 1)a_s \leq R_s\}$. As such, if the situation just described is realized, then:

1.  We are looking for a virtual network segment with the lowest level of overload: $s^* \in S : w_{s^*} = \min\{w_r, r \in S\}$;

2.    In the $s^*$-th virtual network segment, we forcibly terminate an arbitrary active terminal–segment information interaction session: $(m_1, m_{2,}, \ldots, m_{s^*} - 1, \ldots, m_S)$;

3.    We repeat steps 1 and 2 until the condition $C - \sum_{i=1}^{S} m_i a_i \geq a_s$ is satisfied.

Note that performing step 1 of the functional algorithm found not one virtual network segment with the lowest level of overload, but several; the choice of working among them for the further implementation of the operational algorithm is random.

### 3.4. Mathematical Modeling of the System Resource Management Mechanism in the E-Commerce Oriented Ecosystem of the 5Ge Base Station

Suppose that there are enough free system resources in the e-commerce-oriented ecosystem of the 5Ge base station. In that case, it is advisable to meet the needs of virtual network segments at the maximum limits $b_s$: $X_0 = \left[ (m_1, m_2, \ldots, m_s) : \sum_{s=1}^{S} m_s b_s \leq C \right]$.

If the amount of free system resources is limited: $X_1 = X/X_0$, then for their effective distribution between virtual network segments, it is necessary to solve the optimization task of the form:

$$\sum_{s \in S} w_s(m_s) m_s U_s(r_s) \to \max, \tag{4}$$

$$\sum_{s \in S} m_s r_s = C, \tag{5}$$

where $U_s(r_s) = \ln(r_s)$ is the utility function of the parameter of the speed of information interaction for the $s$-th virtual network segment. To solve such a nonlinear optimization task, we apply the approximate gradient projection method [41,42], in which the matrix of projection on the hyperplane (5) is generally defined as

$$P = \left\{ r \in R^S : a_s \leq b_s, s \in S \right\}. \tag{6}$$

In the method chosen for the solution, the iterative procedure for finding the extremum is determined by the expressions:

$$d_k = P \nabla f(x_k), \tag{7}$$

$$x_{k+1} = x_k + \tau_k d_k, \tag{8}$$

$$\tau_k > 0 : x_{k+1} \in P, \tag{9}$$

where x is the vector of controlled variables, and the projection matrix P is defined as

$$P = I - m \left( mm^T \right)^{-1} m = 1 - \frac{1}{\sum_{s \in S} m_s^2} m^T m,$$

where $m = (m_1, m_2, \ldots, m_S)$ is the characteristic vector of the current state of the studied ecosystem. The gradient of the utility function $\nabla f(x_k)$ in Expression (7) is a vector column of the form:

$$\nabla f(x_k) = \left( \frac{w_s m_s}{r_s} \right), s \in S. \tag{10}$$

The value of step $\tau_k$ is chosen so as not to go beyond the range of the domain of admissible solutions $P$, which is determined by the direct constraints (5) and (6) of the optimization task (4). As the initiating values of the elements of the vector $x_0$, we can take the coordinates of the hyperplane intersection point (5), with the diagonal connecting the

vertices $(a_1, a_2, \ldots, a_S)$ and $(b_1, b_2, \ldots, b_S)$ of the parallelogram $P$. The required coordinates will be found as a result of solving a system of linear equations of the form

$$
\begin{cases}
(b_S - a_S)x_1 - (b_1 - a_1)x_S = a_1 b_S - a_S b_1, \\
(b_S - a_S)x_2 - (b_2 - a_2)x_S = a_2 b_S - a_S b_2, \\
\qquad \ldots \\
(b_S - a_S)x_{S-1} - (b_{S-1} - a_{S-1})x_S = a_{S-1} b_S - a_S b_{S-1}, \\
m_1 x_1 + m_2 x_2 + \ldots + m_S x_S = C.
\end{cases}
\tag{11}
$$

The implementation of the iterative procedure (7)–(9) provides movement by the hyperplane (5) from result define by solving the system of linear equations (11) the starting point $x_0$ in the direction of increasing the value of the gradient of the utility function (10) to the point of intersection with the boundary of the domain of admissible solutions $P$. This point where the coordinates intersect will provide the maximum allowable value of the objective function (4).

## 4. Results

To check the adequacy of the mathematical models presented in Section 3, we use the possibilities of simulation modeling. The UML activity diagram for the experiment is shown in Figure 2.

At the initialization stage (0), the values of the established parameters of the studied ecosystem of the 5Ge base station are set and are combined into the set $Init = \{S, \lambda_s, \mu_s, a_s, b_s, C, \max\}$, $s = \overline{1, S}$, where max is the maximum number of received input requests. The next stage waits for the event

(1)  If the event is recorded and it is:

(2)  The stage of the system audit, then:

    (2.1)  The information environment of the 5Ge base station is checked for the presence of overloaded virtual network segments. If:

        (2.1.1)  The overloaded virtual network segment is found, then the functional algorithm described in Section 3.3 is implemented, upon completion of which the control mechanism described in Section 3.4 redistributes the system resources and the system returns to stage (2);

        (2.1.2)  If no overloaded virtual network segment is found, then the system returns to the waiting stage for the event (1);

(3)  The stage receives a new incoming request to the target virtual network segment, then:

    (3.1)  The availability of the required amount of free system resources should be checked. If:

        (3.1.1)  The amount of free system resources is sufficient, then the incoming request is received, the redistribution of system resources between received incoming requests, which is controlled by the control mechanism described in Section 3.4, is carried out, and the system returns to the stage where it is waiting for the event (1);

        (3.1.2)  If the amount of free system resources is not enough, then it is checked whether there is an overloaded virtual network segment in the information environment of the 5Ge base station. If:

            (3.1.2.1)  There is no overloaded virtual network segment, then the input request is lost, and the system returns to stage (1);

            (3.1.2.2)  If the overloaded virtual network segment is found, then the functional algorithm described in Section 3.3 is implemented. Then, the control mechanism described in Section 3.4 redistributes system resources, and the system returns to stage (3.1).

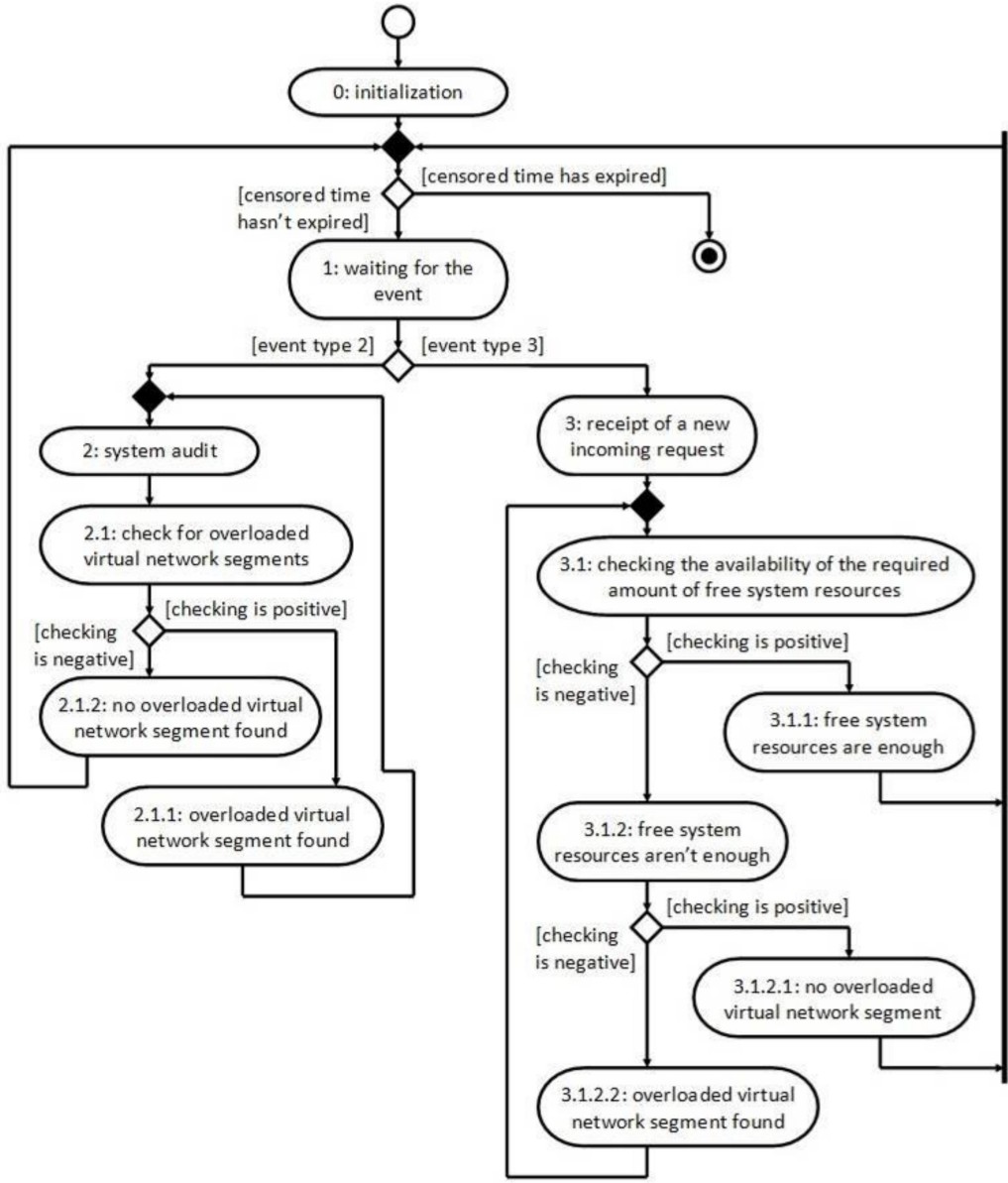

**Figure 2.** The UML activity diagram of the experiment on the simulation of the operation of the 5Ge base station, the information environment of which supports the process of virtual network segments.

The computational experiments were performed in the 5Ge ecosystem (LTE Advanced Pro) 800 MHz band. The actual minimum and maximum connection speeds of $a_s = 0.75$ Mbps and $a_s = 80.00$ Mbps were empirically determined for the Huawei equipment that was used. The system was studied in a metropolis, where the distance from the base station to the final mobile terminal ranged from 4 to 13 km. The upper limit of information communication speed for all of the active terminal–segment interaction sessions was selected from the range of available values: $C = \{325, 345, 385, 405, 425, 445\}$ Mbps. The system resources limited by the parameter's $C$ value was distributed between three virtual network segments $(S = 3)$, for which the following guaranteed amounts of available system resources were set: $\overline{R} = \{225, 150, 75\}$ Mbps. The intensities of the receipt and service of the incoming requests were equal for all of the virtual network segments and were: $\lambda = \{30, 30, 30\}$, $\mu = \{30, 30, 30\}$.

As a result of the experiments, the functional dependencies $D_s = f(C)$, $\overline{R}_s = f(C)$, $\Delta T_s = f(C)$, $s = \{1, 2, 3\}$ were obtained, where $D_s$ is the probability of losing the input request addressed to the $s$-th virtual network segment, %; $\overline{R}_s$ is the average actual volume

of the *s*-th virtual network segment, c.u.; $\Delta T_s$ is the total relative (to the entire censored period of operation of the studied system) duration that the *s*-th virtual network segment was in the overload state, %; and *C* is all of the available system resources that are available for distribution, c.u. Empirically obtained graphs of the functions $D_s = f(C)$, $\overline{R}_s = f(C)$, $\Delta T_s = f(C)$, $s = \overline{1,3}$, are presented in Figures 3–5, respectively.

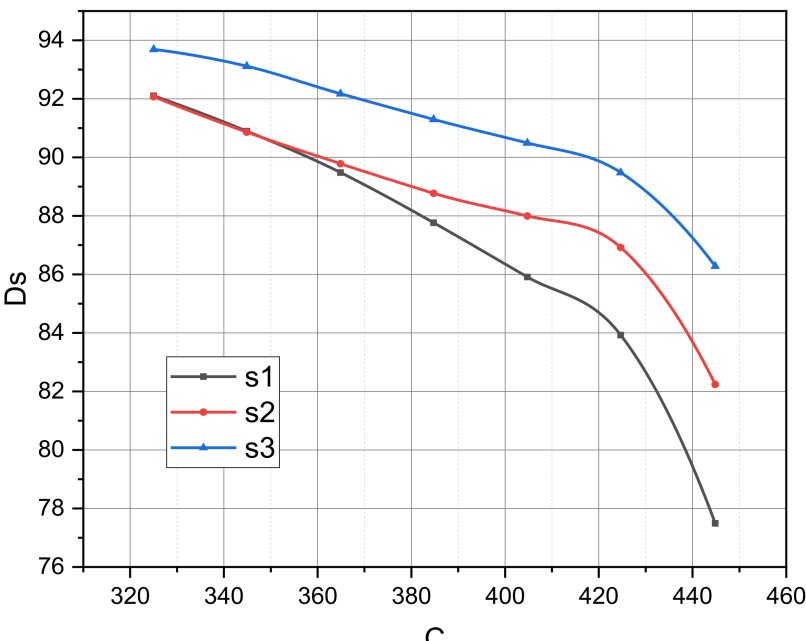

**Figure 3.** Functional dependence $D_s = f(C)$ for the studied ecosystem of the 5Ge base station.

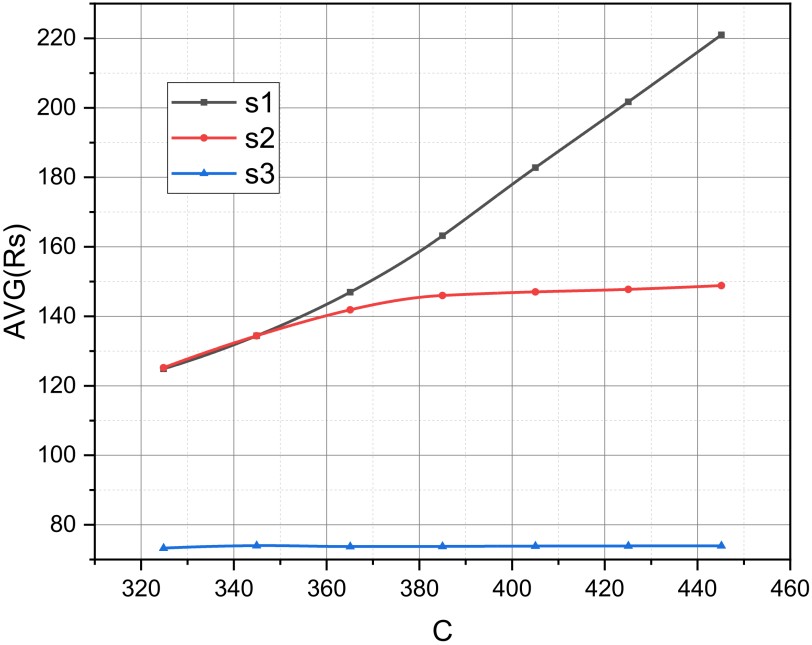

**Figure 4.** Functional dependence $\overline{R}_s = f(C)$ for the studied ecosystem of the 5Ge base station.

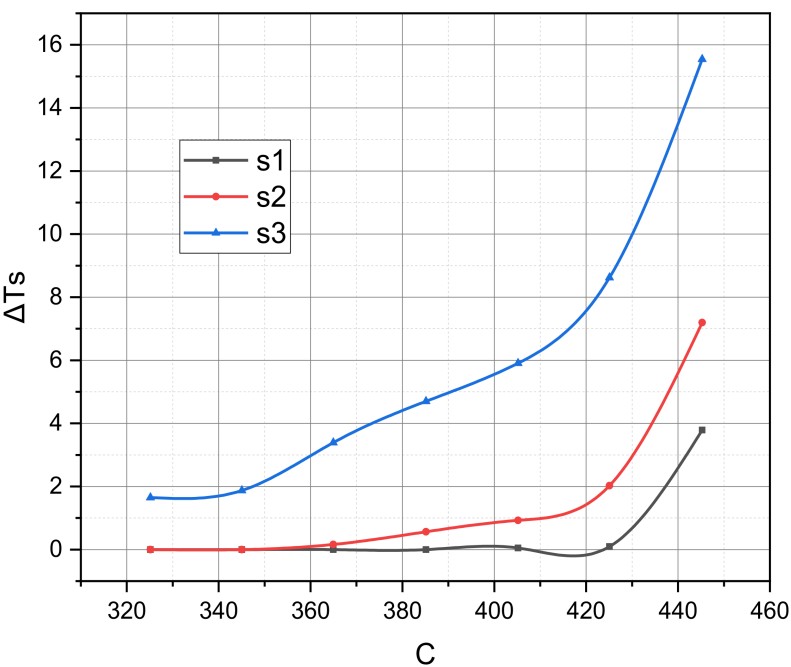

**Figure 5.** Functional dependence $\Delta T_s = f(C)$ for the studied ecosystem of the 5Ge base station.

The calculations visualized in Figures 3–5 were performed in the software environment OriginPro 2018 (64 bit) SR1.

## 5. Discussion

Analyzing the first approximation of the graphs of functions $D_s = f(C)$ shown in Figure 3 for three virtual network segments $(s_1, s_2, s_3)$, which functioned for a censored time in the information environment of the studied ecosystem of the 5Ge base station, we can state that the probability of the loss of input requests decreases with increasing maximum available amounts of system resources. However, considering that the degree of a load of all virtual network segments was the same (the intensity of receipt and service of incoming requests were equal: $\lambda = \{30, 30, 30\}$, $\mu = \{30, 30, 30\}$), but the distribution of guaranteed amounts of available system resources between segments was uneven ($\overline{R} = \{225, 150, 75\}$ Mbps), then the initial value and rate of decrease in the graphs of the functions $D_s = f(C)$ with the increasing importance of the argument turned out to be different. The connection is obvious—the greater the guaranteed amount of available system resources for a particular virtual network segment, the lower the inherent probability of losing the incoming request was. The nonlinear nature of the graphs of the functions shown in Figure 3 correlates with the analytical representation of the mechanism to control the allocation of system resources in the e-commerce-oriented ecosystem of the 5Ge base station (see Section 3.4). The underlying optimization task with objective function (4) and constraints (5) and (6) was classified as nonlinear. At the same time, the descending nature of the graphs of the functions $D_s = f(C)$, $s = \overline{1,3}$, unambiguously confirms the effectiveness of the operation of the just mentioned control mechanism.

The graphs of functions $\overline{R}_s = f(C)$, $s = \overline{1,3}$, which are presented in Figure 4 in the order of dynamics of increasing their values by increasing value of the argument, can be arranged as follows: $\overline{R}_3 = f(C)$, $\overline{R}_2 = f(C)$, $\overline{R}_1 = f(C)$. Again, this order correlates with the initial distribution of guaranteed amounts of available system resources between virtual network segments ($\overline{R} = \{225, 150, 75\}$ Mbps). The example of graphs of functions $\overline{R}_3 = f(C)$ $\overline{R}_2 = f(C)$ clearly shows the operation of the control mechanism for the allocation of system resources. It does not allow the virtual network segments to be overloaded, even in the conditions of a steady stream receiving new incoming requests.

This is the responsibility of the system audit service and is based on the functional algorithm of the forced termination of the active session of terminal-segment information interaction for the overloaded virtual network segment (see Section 3.3).

Finally, the growing nature of the graphs of the functions $\Delta T_s = f(C), s = \overline{1,3}$ from Figure 5 indicates the need for this algorithm. Recall that $\Delta T_s$ is the total relative (to the entire censored period of operation of the studied system) duration that the $s$-th virtual network segment was in the overload state, %. As the value of the argument, i.e., the total amount of system resource involved, increases and as the intensities of new incoming requests and the processing of received incoming requests are the same for all virtual network segments, there is a tendency to overload them rapidly. Moreover, the third segment $(\Delta T_3 = f(C))$ suffers the most because of the primary guaranteed amount of allocated system resources, the smallest being $(\overline{R}_3 = 75 \text{ Mbps})$. However, let us compare the graphs of the functions shown in Figures 3 and 5. It becomes evident that despite the progressive overload with the growth of the argument $C$, the e-commerce-oriented ecosystem of the 5Ge base station continues to fulfill its functional purpose. This fact empirically proves the adequacy of the profile of the mathematical model given in Section 3.2 and the functionality and efficiency of the algorithms presented in Sections 3.3 and 3.4.

## 6. Conclusions

At the end of 2020, the number of Internet users exceeded the $4.5 \cdot 10^9$. If we talk about the balance between stationary and mobile means of e-commerce information interaction, the boundary between these poles is increasingly mixed in the direction of the latter. The traffic consumed by user terminals is also overgrowing. All of these trends prove the need for the evolution of mobile communication generations, in particular, the transition from 4G to 5G. According to the specifications, the latter should be implemented into flexible services for traffic management, improving the efficiency of which will remain an actual task.

For the first time, the current article proposed a mathematical model for the operation process of an e-commerce-oriented ecosystem of a 5Ge base station, the information environment of which supports the operation of independent virtual network segments that provide terminal–segment information interaction services. In contrast to the existing ones, the presented model describes the studied process as a multi-pipeline queuing system, the inputs of which are coordinated with the flows of requests for communication with the relevant virtual network segments. The distribution of total resources between weighted virtual network segments in the simulated system was conducted dynamically by the appropriate software control mechanism. It considers the address intensities of new incoming requests and the maintenance of received incoming requests but throughout the scale of the information environment of the 5Ge base station ecosystem. Based on the created mathematical model, the functional algorithm for the forced termination of an active terminal–segment information interaction session in an overloaded virtual network segment and the control mechanism of the distribution of the released system resources between other virtual network segments were formulated by taking the degree of their overload into account.

The simulation and computational experiments showed that the implemented forced termination algorithm and system resource management mechanism allowed the 5Ge base station to continue receiving incoming requests despite the overload of individual virtual network segments. It was empirically shown that the proposed services were effectively scaled concerning the value of the system resources that were generally available for allocation and the allocation method within which the guaranteed amounts of system resources for individual virtual network segments were.

Further research will focus on the formalization of qualitative metrics based on the values of the parameters, from which it will be possible to order the activation process of the system audit.

**Author Contributions:** Conceptualization, V.K.; methodology, V.K.; software V.K.; validation, I.I.; formal analysis, V.K.; investigation, V.K.; writing—original draft preparation, V.K.; writing—review and editing, I.I.; supervision, V.K. All authors have read and agreed to the published version of the manuscript.

**Funding:** This research received no funding.

**Institutional Review Board Statement:** Not applicable.

**Informed Consent Statement:** Informed consent was obtained from all subjects involved in the study.

**Data Availability Statement:** Most data are contained within the article. All of the information is available upon request due to restrictions, e.g., privacy or ethics.

**Conflicts of Interest:** The authors declare no conflict of interest.

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
