# Peer review of "Study of the Operation Process of the E-Commerce Oriented Ecosystem of 5Ge Base Station, Which Supports the Functioning of Independent Virtual Network Segments"

_jtaer, doi:10.3390/jtaer16070158_

Round 1

Reviewer 1 Report

1. The main objectives of the research are defined at the introduction of the study. The authors described the study problem and research questions, the importance of the study, and the hypotheses as well.
2. The state of art/Literature review covers the most important and relevant international literature sources in an appropriate structure. The literature sources are highly acceptable and most of the relevant literature sources are used, the in-text citations are used well, citation style is correct.
3. All the tables and figures are clear, understandable, and relevant, sources are indicated in each case well.
4. The authors have completed the necessary evaluations. Conclusions and recommendations are well structured, those are in relevance with the analysis and discussion. Conclusions are suitable for gaining new results and initiating further or new research. The new results are drawn up in an understandable way.
5. In materials and methods are good and comprehensive overview of the topic, based on a wide range of literature. The methodological contains a correct description of the methods applied, are well documented and supported.
6. The conclusions show that the authors have good and deep knowledge of the topic. Novel findings and recommendations are well articulated.
I accept this study because it is very current and important. Therefore, with minor editing and format of the text, by use of a professional English proofreader, the limitations and further areas of research can be more highlighted, and the manuscript will be appropriate and ready for publication.

Author Response

Dear Sir/Madam,

we believe that only a real professional and scientist can appreciate someone else's research. Thank you for your high praise of our paper. We have very carefully re-read the paper and improved the formatting and English mistakes. In addition we have reorganised and extended Introduction section. We have added the motivation of this paper, clear point-by-point the main contribution and the remainder of this paper.

Reviewer 2 Report

The work conducts research in the process of operation of the e-commerce oriented ecosystem of the base station 5Ge, the information environment of which supports the functioning of independent virtual network segments. The simulation and computational experiment showed that the implemented forced termination algorithm and system resource management mechanism allow the 5Ge base station to continue receiving new incoming requests despite the overload of individual virtual network segments. 

The topic fits very well the scope of the journal. 

The manuscript is well written, the structure of the paper is clear and the language is proper.

The introduction section should be reorganised in order to clarify the motivation, objectives (aligning with the section 3.1) and the contributions in which advances the related work.

The contributions should be well delimited in the introduction section in order to advance the state of the art. It is not clear the real contributions compared to related work.

The last paragraph of section 1 should write the organisation and structure of the manuscript.

Authors should better justify the default simulation factor/parameters used and also the scenario simulated compared to the real environment. Additionally, which simulator was used? Is it a developed one? If yes, authors should describe in detail the implementation.

Authors says: "... It is empirically shown that the proposed services are effectively scaled in relation ….". How to measure this empirically? Please, clarify.  

Section 5 should be improved in order to describe more details about the results obtained mainly comparing/discussing with the related work.

The manuscript needs a broad revision in order to correct English and typos.

Author Response

Dear Sir/Madam,, thank you for the time spent on our paper and the appreciation of our research.In response to your recommendations:

  1. «The introduction section should be reorganised in order to clarify the motivation, objectives (aligning with the section 3.1) and the contributions in which advances the related work.»

Thank you for your suggestions. We have reorganised and extended Introduction section. We have added the motivation of this paper, point-by-point the main contribution and the remainder of this paper

  1. «The contributions should be well delimited in the introduction section in order to advance the state of the art. It is not clear the real contributions compared to related work.»

Thank you for this valuable comment. We have added such a paragraph to Section 1:

“The main contributions of this paper can be summarized as follows:

  1. we have created a mathematical model for the operation process of the e-commerce oriented ecosystem of 5Ge base station, the information environment of which supports the operation of independent virtual network segments that provide terminal-segment information interaction services;
  2. we have designed the functional algorithm of forced termination of an active session of terminal-segment information interaction in the overloaded virtual network segment;
  3. we have formulated the mechanism of control of the distribution of the released system resources between other virtual network segments taking into account the degree of their overload;
  4. we made the simulation and computational experiment showing that the implemented forced termination algorithm and system resource management mechanism allow the 5Ge base station to continue receiving incoming requests despite the overload of individual virtual network segments.»

In addition, we have added the remainder of this paper at the end of Section 1:

“The structure of the article is as follows: 1. The second section provides a critical analysis of the state of research related to the topic of the article. The results of the analysis revealed an area that has so far been ignored in known studies. The subject of the research was also selected in the identified area; 2. In the third section, the research objectives are set and the proposed information technology for describing the process of information interaction in the e-commerce-oriented ecosystem of the 5Ge base station is theoretically substantiated; 3. The fourth and fifth sections are devoted to simulation modeling of the presented information technology and analysis of the adequacy of the results demonstrated by it; 4. In the sixth section the conclusions concerning urgency of the chosen theme and the received theoretical and applied results are formulated. The direction of further research is also determined.”

  1. «Authors should better justify the default simulation factor/parameters used and also the scenario simulated compared to the real environment. Additionally, which simulator was used? Is it a developed one? If yes, authors should describe in detail the implementation.»

Thank you for your comment. All the initial data that allowed to calculate the functional dependencies shown in Fig. 3-5 are given in the fourth section. The adequacy of the simulation is ensured by the correctness of the mathematical apparatus presented in section three and the content of the algorithm presented in Fig. 2. Direct calculations were performed in the software environment OriginPro 2018 (64 bit) SR1. Before Fig. 3, the phrase "Calculations visualized in Figures 3-5, performed in the software environment OriginPro 2018 (64 bit) SR1" have been added.

  1. «Authors says: "... It is empirically shown that the proposed services are effectively scaled in relation ….". How to measure this empirically? Please, clarify.»

Thank you for your comment. We have used the word “empirical” as a synonym for “experimental”. This phrase summarizes the content of the Section 5 of the paper.

  1. «Section 5 should be improved in order to describe more details about the results obtained mainly comparing/discussing with the related work.»

Thank you for your comment. As already mentioned, the main scientific result presented in the article is the models and methods presented in Sections 3 and 4. Their features and differences from analogues are formulated in the second paragraph of the conclusions. The conclusions are formulated in such a way as to generalize as compactly as possible the results that correspond to the research objectives that were formulated in Part 3.1.

7. The manuscript needs a broad revision in order to correct English and typos.

Thank you for your suggestion. We have very carefully re-read the paper and improved the English and typos. 

Best wishes, Authors.
